# Counting young birds: A simple tool for the determination of avian population parameters

Werner Oldekop[1], Gerd Oldekop[1], Kai Vahldiek[2], Frank Klawonn[2], Ursula Rinas[1]*

1 Institute of Technical Chemistry/Life Science, Leibniz University of Hannover, Hannover, Lower Saxony, Germany, 2 Institute for Applied Informatics, Ostfalia University of Applied Sciences, Wolfenbüttel, Lower Saxony, Germany

* rinas@iftc.uni-hannover.de

**Data Availability Statement:** All relevant data are within the paper and its Supporting information files.

## Abstract

Population parameters are usually determined from mark-recapture experiments requiring laborious field work. Here, we present a model-based approach that can be applied for the determination of avian population parameters such as average individual life expectancy, average age in the population, and generation length from age-differentiated bird counts. Moreover, the method presented can also create age-specific results from lifetime averages using a deterministic exponential function for the calculation of parameters of interest such as age-dependent mortality and age distribution in the population. The major prerequisites for application of this method are that young and adult birds are easily distinguishable in the field as well as the existence of sufficiently large data sets for error minimization. Large data sets are nowadays often available through the existence of so-called "citizen science" data-bases. Examples for the determination of population parameters are given for long-living migratory birds which travel as families in large groups such as the Common Crane and the Whooper Swan. Other examples include long-living partially migratory birds staying together in large flocks which do not travel as families such as the Black-headed Gull, and also short-living songbirds where at least from one sex young and adult birds are easily differentiable such as the male Black Redstart.

## Introduction

Nowadays, the mark-recapture method is the most commonly used method to determine population parameters such as abundance, mortality, survival, and others. The method is based on marking animals and their re-encounter at a later time [1]. This method is continuously refined (e.g. [2], http://www.phidot.org/software/mark/) and is nowadays not only applied to study avian population dynamics but also to study any animal—even small ones such as insects—if species-adapted markings are available. Although this method is considered as the gold standard it has also some drawbacks. Most obviously, it depends on animal markings and their recaptures/resightings, thus, requiring laborious and specialized field work. For example,

**Funding:** The authors received no specific funding for this work.

**Competing interests:** The authors have declared that no competing interests exist.

bird marking by ringing or tagging is an intrusive process and requires permissions and dedicated trained persons.

Here we present a simple method that allows the determination of population parameters such as the average individual life expectancy, **L**, the average age in the population, **A**, and the generation length (defined here as the average age of the breeding population), **G**, from counts of young and adult birds.

Moreover, among others the following age-dependent variables can be determined:

1. the age-dependent mortality, i.e. the probability of dying at a certain age, **M(a)**

2. the age distribution in the population, **P(a)**

3. the remaining life expectancy at a certain age, **Le(a)**

4. and the relative mortality, i.e. the probability of dying in the following year (a+1) when a certain age has been reached, **rM(a)**

The main prerequisites of this method encompass that young and adult birds are easily distinguishable in the field and the existence of large data sets. The latter is nowadays often available through citizen science databases. The central input data into the calculations are the portion of young or juvenile birds in the population, **g = juv/juv+ad**. In the following sections we shortly introduce the mathematical background of the method, present some examples and discuss the pro and cons of this approach. The details of the mathematics are given in the Supporting Information.

## Mathematical background

The basic concept of this method is simple and best explained by imaging a huge water tank, e.g. containing a water volume of 100 m$^3$ with green and red colored particles (corresponding to the bird population of young (green) and adult birds (red), respectively) into which there is an annual inflow of 20 m$^3$ fresh water on top containing only green particles. With time green particles turn into red particles in the tank, i.e. young birds become adult birds. If the same amount of particle containing water (here 20 m$^3$) is removed each year from the bottom of the tank (corresponding to the portion of birds which die) and if the particle concentration in the inflow (green particles) is identical to the particle concentration in the tank (green and red particles) then we have a situation which mimics a state where the mortality rate equals the birth rate i.e. there is no population growth or population decline. If there would be no mixing in the tank we would have in the example given above with 20% annual inflow a portion of young (green) "birds" of **g = 0.2** and all "birds" would die at the age of 5 years. With mixing the average individual life expectancy **L** is also 5 years, but some "birds" stay longer in the tank, i.e. die at later age, and some shorter, i.e. die younger.

Thus, in a constant population with a constant ratio of young birds the average individual life expectancy can be—independent of all other parameters—calculated as follows:

$$L = 1/g \tag{1}$$

With this basic input information and a mathematical model mainly based on mass balances, population parameters can be determined such as the age-dependent mortality i.e. the probability of dying at a given age, the age distribution within the population, the remaining life expectancy at a given age and the relative mortality i.e. the probability of dying when a certain age has been reached. Details of the mathematical background are described in the S1 File. Moreover, explanations and tools to calculate population parameters including corresponding

confidence intervals using Excel or R are also given in the Supporting Information (Excel: S1 and S2 Files; R: S1 File, S3 and S4 Files).

## Examples

In the following section we want to apply the described method to determine population parameters for a set of different bird types including long-living migratory birds which travel as families (Common Crane and Whooper Swan), long-living partially migratory birds where young birds tend to separate from the adults (Black-headed Gull), and also short-living (partially migratory) songbirds (Black Redstart). In all these examples birds in their first year are easily distinguishable from adult birds (Fig 1).

Thus, age-differentiated bird counts can be taken e.g. directly after fledging and also later in time. This is important as the mortality of very young birds in their first weeks or months is higher than during later stages of their life and thus, data sets including hatchlings, fledglings or very young birds have to be considered differently for the determination of these parameters. Examples presented here relate to birds which already survived the first most vulnerable part in their life. However, for the last example, the Black Redstart, we will also consider the higher mortality during the first part of a bird's life. All calculations shown for the following examples are based on published data or the database *ornitho.de*.

### Common Crane (*Grus grus*)

The Common Crane is a migratory bird [6]. The main breeding regions encompass the Scandinavian countries Sweden and Finland as well as Eastern Europe and Russia. Birds from the European breeding range spend the winter mainly in the Iberian Peninsula where families stay together at least until spring migration. Young birds are easily distinguished from the adults in the field; adult birds have distinctive facial markings while young birds have a plain brownish face (Fig 1A). The portion of young birds in the population can be nicely determined during autumn migration when Common Cranes pause in huge numbers at traditional stopover places, e.g. in Mecklenburg-Vorpommern, Germany. Birds resting in autumn in Mecklenburg-Vorpommern originate mainly from Fennoscandian breeding regions [7] where breeding starts in late April and hatching around one month later [8]. This breeding population is still slightly increasing in size [6]. For the determination of the portion of young cranes, data collected during the main stopover month October from 2012 until 2020 deposited at the database *ornitho.de* were used (data set and calculations in Excel accessible in S2 File, calculations in R can be performed using S3 and S4 Files, for detailed explanations see S1 File). Employing the modified exponential mortality function as described in the S1 File with the following input variables: portion of young birds g = 0.095, population growth r = 0.02 (2% per year), maximum age $a_{max}$ = 30 years (oldest bird determined from ringing data as 24 years and 3 months [9]) the following population parameters were identified:

| | |
|---|---|
| average life expectancy, L | 12.3 years (from October) |
| average age in the population, A | 7.6 years (in October) |
| generation length, G | approx. 11.5 years (first breeding with 5 years, [8]) |

Please have in mind that the average life expectancy relates to a bird that already survived the time until his first autumn migration. Moreover, the age-dependent mortality i.e. the probability of dying at a given age, the age distribution within the October population, the remaining life expectancy at a given age and the relative mortality i.e. the probability of dying when a certain age has been reached are determined (Fig 2) based on the model specified in S1 File.

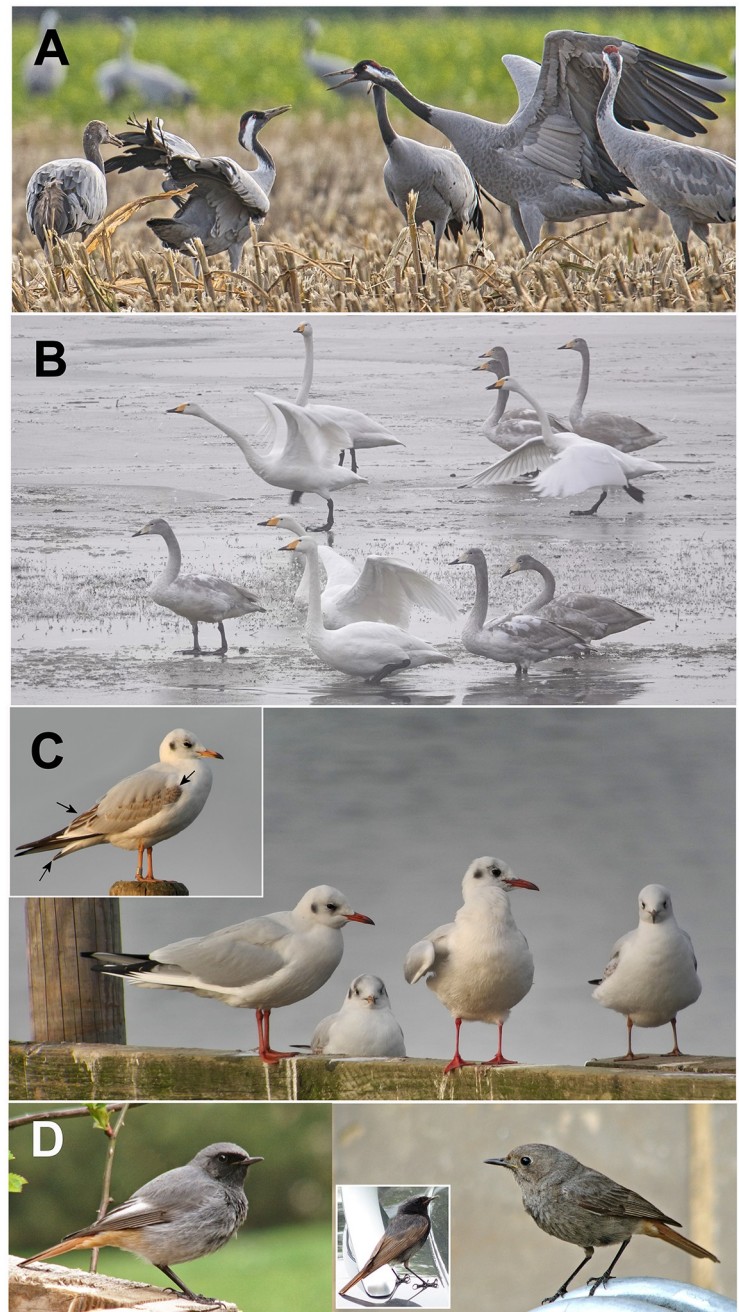

**Fig 1. Plumage of young and adult birds during census time. (A) Common Crane:** In the foreground four adult birds with distinctive facial markings and a young bird at the left side with a plain brownish face (Photo: Frank Hessing, Wietingsmoor, Germany, 30.10.2021). **(B) Whooper Swan:** Five adult birds with white plumage and six young birds with a grayish plumage and paler yellow beaks (Photo: Ursula Rinas, Polder Lenzen, Germany, 20.01.2019). **(C) Black-headed Gull:** Main image with two adult and two young birds. Adult birds have bright red beaks and legs. Young birds have more orange beaks and legs and remaining brownish juvenile coverts and tertials and a black tail band. The insert shows a side view of a young bird. Arrows are pointing to the differences of young bird plumage (Photos: Ursula Rinas, Steinhude, Deutschland, 12.11.2016). **(D) Black Redstart:** The left-hand image displays a contrast rich adult male with a prominent white wing panel and the right-hand image a young male (approx. one year old) with a duller female-like coloration (cairii plumage type, [3–5]). The insert in the right image shows a young male of the more advanced paradoxus-plumage type [3–5] (Photos: Bernd Nicolai, Halberstadt, left: 21.04.2006, right; 21.07.2005, insert: 12.07.2015).

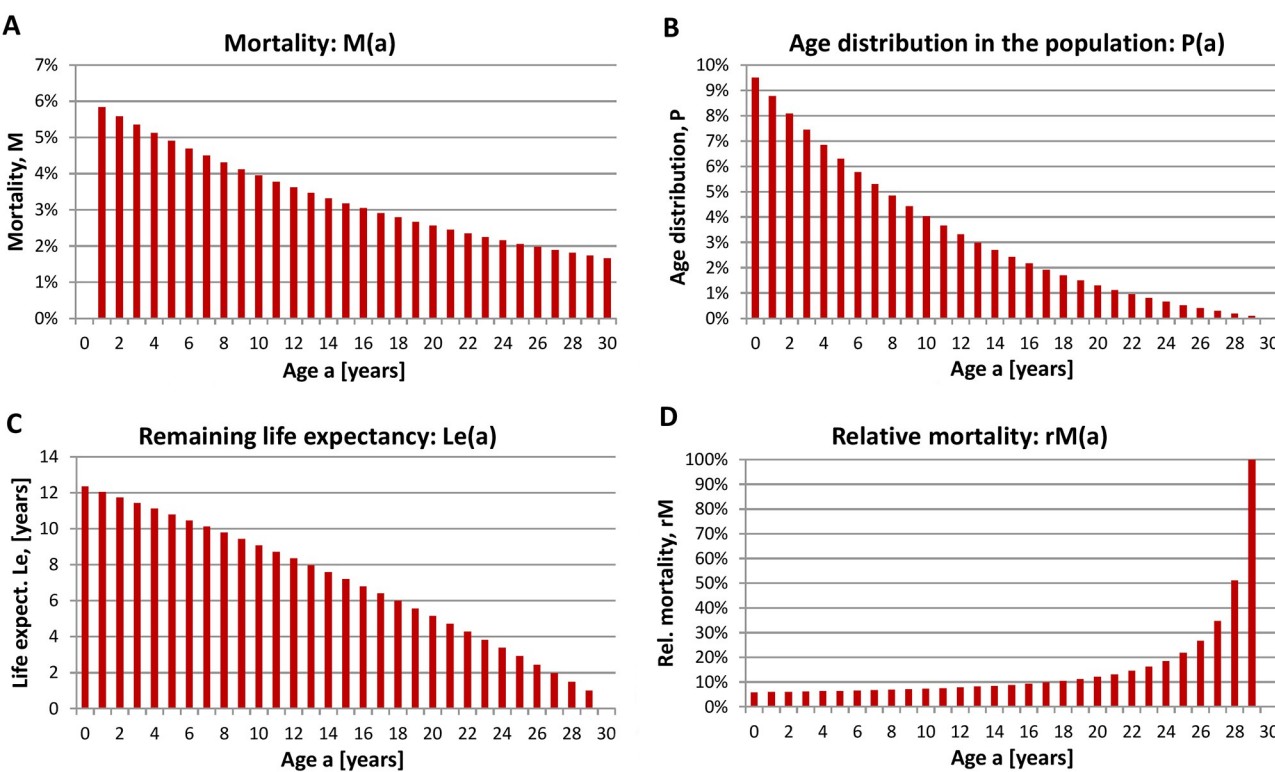

**Fig 2. Population dynamics of Common Cranes. (A)** Mortality, M(a) (probability of dying at a given age), **(B)** age distribution in the October population P(a), **(C)** remaining life expectancy at a given age Le(a), and **(D)** relative mortality rM(a) (probability of dying when a certain age has been reached) of the Common Crane. Calculations are based on age-differentiated bird counts determined during October in Germany (2012–2020) northeast of the geographic coordinates 12˚00 E, 53˚00 within Germany (mainly Mecklenburg-Vorpommern and the northern part of Brandenburg). Raw data are from the database *ornitho.de*.

## Whooper Swan (*Cygnus cygnus*)

The Whooper Swan is also a migratory bird with a distinct breeding and wintering range, e.g. the majority of birds wintering in Germany are known to breed in Northern and Eastern Europe as well as in Western Siberia [7] where breeding starts in May and hatching around one month later [8]. Whooper swans also stay together as families in the wintering area and the young swans can be easily distinguished during this time from the adults as young swans have a more grayish plumage and their beaks are more pale yellow (Fig 1B). In Germany wintering Whooper Swans reach their highest numbers in January/February [7, 10] Thus, population parameters were determined from the age-differentiated counts of Whooper Swans in January (2012–2020) available through the database *ornitho.de* (data set and calculations in Excel accessible in S2 File, calculations in R can be performed using S3 and S4 Files, for detailed explanations see S1 File). Employing the modified exponential mortality function as described in S1 File with the following input variables (portion of young birds g = 0.17, population growth r = 0, maximum age $a_{max}$ = 30 years (maximum age determined through ringing data as 26 years and 6 months [9]) the following population parameters were identified:

| | |
|---|---|
| average life expectancy, L | 5.9 years (from January) |
| average age in the population, A | 4.7 years (in January) |
| generation length, G | approx. 9 years (first breeding with 5 years, [8]) |

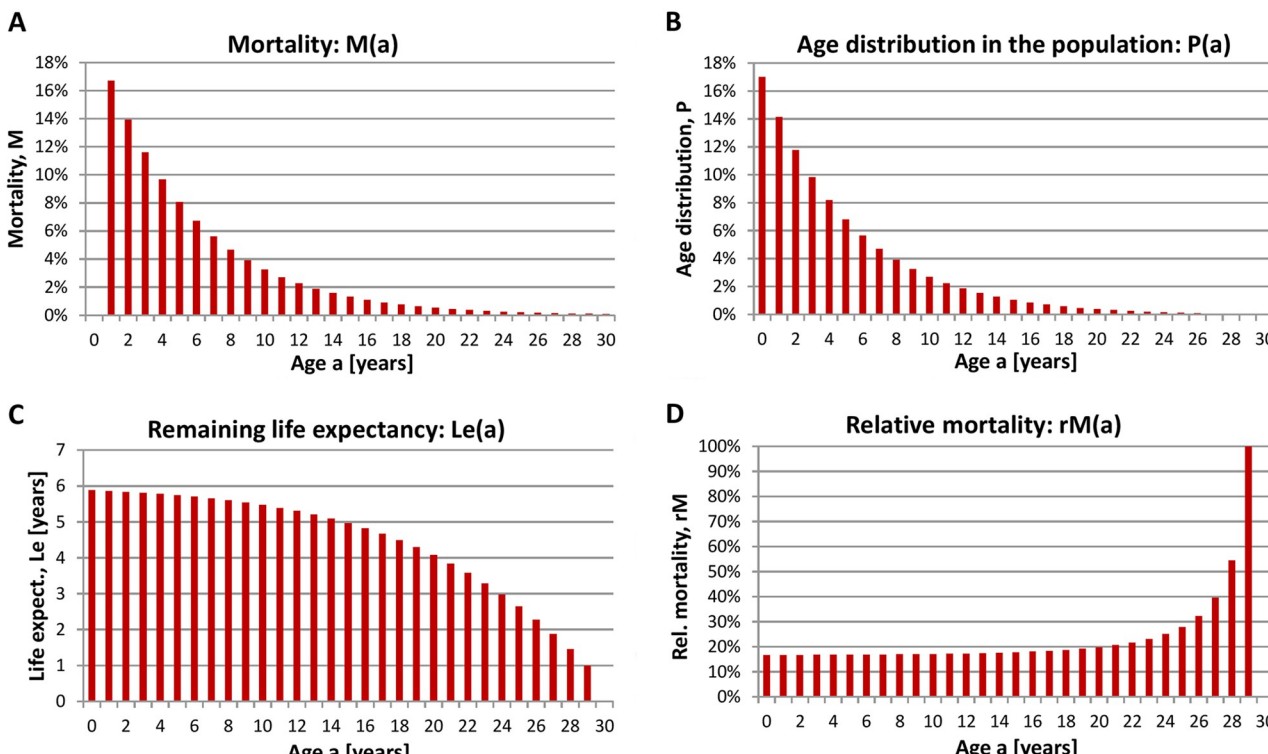

**Fig 3. Population dynamics of Whooper Swans.** (**A**) Mortality, $\underline{M(a)}$ (probability of dying at a given age), (**B**) age distribution in the January population $\underline{P(a)}$, (**C**) remaining life expectancy at a given age $\underline{Le(a)}$, and (**D**) relative mortality $\underline{rM(a)}$ (probability of dying when a certain age has been reached) of the Whooper Swan. Calculations are based on age-differentiated bird counts determined during January in Germany (2012–2020, whole country of Germany). Raw data are from the database *ornitho.de*.

Again, it should be noted that the average life expectancy relates to a bird that already survived until its first mid-winter. In addition, the age-dependent mortality i.e. the probability of dying at a given age, the age distribution within the January population, the remaining life expectancy at a given age and the relative mortality i.e. the probability of dying when a certain age has been reached are shown (Fig 3).

## Black-headed Gull (*Chroicocephalus ridibundus* syn. *Larus ridibundus*)

Black-headed Gulls are found year-round in Germany, however, in winter breeding birds from Germany move south-west and birds wintering in Germany originate from north-eastern countries [7]. Young and adult Black-headed Gulls are easily distinguishable in the field close to the end of their first year (Fig 1C, [11]).

Different from the examples shown above young and adult birds do not stay together as families. Moreover, young and adult birds even tend to segregate. This segregation has been interpreted in such a way that younger birds gather together to avoid competing with the older birds, which are e.g. more experienced in catching food [12]. Earlier studies revealed that this particular behavior requires a sufficiently large data set for reliable determination of population parameters as well as the appropriate time in the year for the bird counts [11]. During the first two months after fledging, July and August, the portion of young birds is considerable higher in the population compared to the remaining time until the next breeding season revealing the higher mortality in the first period of life [11]. Moreover, during July and August

Black-headed Gulls also strongly tend to segregate into clusters of different age groups [11]. Thus, it is recommended to utilize the data from age-differentiated bird counts later on in the year when the groups show a more even age distribution. Finally, the amount of age-differentiated bird counts for the Black-headed Gull is considerably lower compared to the available data for the Common Crane and the Whooper Swan, thus data should be utilized from a more extended time period. Considering the above points bird counts were taken from September until November (2012–2020) available through the database *ornitho.de* (data set and calculations in Excel accessible in S2 File, calculations in R can be performed using S3 and S4 Files). By employing the modified exponential mortality function as described in S1 File with the following input variables (portion of young birds g = 0.2, population growth r = 0, maximum age $a_{max}$ = 30 years (maximum age determined through ringing data as 32 years and 9 months [9]) the following population parameters were identified for those birds which survived the first 2–3 vulnerable months after fledging.

average life expectancy, L          5 years (from autumn)

average age in the population, A     3.9 years (in autumn)

generation length, G                         approx. 7 years (first breeding with 3 years, [8])

Moreover, the age-dependent mortality i.e. the probability of dying at a given age, the age distribution within the autumn population, the remaining life expectancy at a given age and the relative mortality i.e. the probability of dying when a certain age has been reached are given (Fig 4). Again, the average life expectancy as well as the other variables determined for

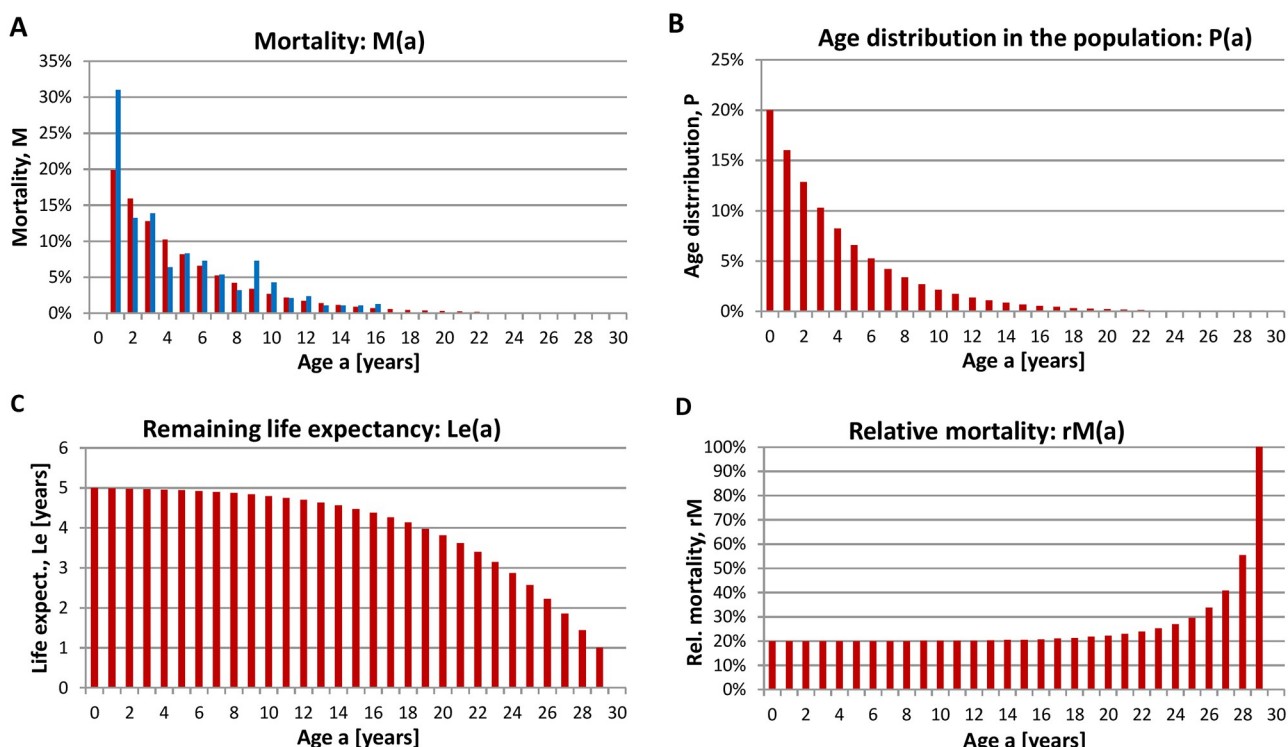

**Fig 4. Population dynamics of Black-headed Gulls. (A)** Mortality, M(a) (probability of dying at a given age) determined using age-differentiated bird counts (red bars, this study) in comparison to mark-recapture data (351 ring recoveries) as determined by Flegg and Cox [13] (blue bars), **(B)** age distribution in the autumn population P(a), **(C)** remaining life expectancy at a given age Le(a), and **(D)** relative mortality rM(a) (probability of dying when a certain age has been reached) of the Black-headed Gull. Calculations are based on age-differentiated bird counts determined from September—November in Germany (2012–2020, whole country of Germany). Raw data are from the database *ornitho.de*.

the Black-Headed Gull as well as for the other two examples given (Common Crane and Whooper Swan) relate to those birds which have survived the first most vulnerable parts in their life after hatching and fledging.

A comparison of the age-dependent mortality of the Black-Headed Gull determined by the method described here or by utilizing mark-recapture data [13] revealed an almost identical profile except for the first year mortality (Fig 4A). The discrepancy for the first year mortality is expectable in this case as we did not consider the high mortality in the first months after hatching. Considering the high first year mortality requires age-differentiated bird counts in different time-segments starting with the first segment after hatching and fledging until age-differentiated bird counts reach constant values [11].

In the following an example is given, a little songbird, where we also consider the risky time directly after fledging.

## Black Redstart (*Phoenicurus ochruros*)

The Black Redstart is a little songbird originally living in rocky mountainous regions but nowadays mainly found in residential areas close to houses [14]. Young and old male Black Redstarts are easily distinguishable from each other in the breeding season as old male Black Redstarts show a white wing panel which the young males hatched in the year before do not show (Fig 1D, [3, 4]). The white wing panel appears the first time in male second calendar year birds (cy2) after the breeding season and their first complete molt in late summer. Moreover, young males appear in two different plumage types with the majority having a female like appearance (cairii plumage-type) and a smaller part carries an advanced so-called paradoxus plumage-type ([3, 4], Fig 1D). Both young and old males show territorial behavior during the breeding season [15] and, thus, the portion of young males can be determined by counting singing birds without (young males) and with a white wing panel (adult males).

The following calculations are based on a data set of counts of young and adult males during the breeding season from Halberstadt (Sachsen-Anhalt, Germany, from 1982 until 2015, [15, 16]). The analysis of these data revealed an average portion of young birds g = 0.49 (~0.5) assuming that males and females occur in equal numbers. The total population can be taken as stable as there was no trend in the data during the period considered.

From this data the average life expectancy L = 1/g can be determined as two years for a bird from his first breeding season onwards translating into an average total life expectancy of 3 years for those birds which survived their first year.

Birds within their first year have on average a much lower life expectancy, in particular within their first weeks and months. An estimation of the average life expectancy in the first year is possible if the average breeding success of the birds is known. For Black Redstarts 6.5 fledglings have been determined per year and breeding pair [15] translating into a portion of fledglings in the post-breeding population of $g_f$ = 0.76 and an average (total) life expectancy for a fledgling of $L_f$ = 1/$g_f$ = 1.3 years. Assuming again equal numbers of fledging males and females, a constant total population and an average portion of 50% last year birds (g = 0.5) in the pre-breeding population, only approx. 15% of the fledglings reach their first breeding season.

Based on these data the average age in the population directly prior to breeding (pre-breeding population) can be determined as 2 years and the generation length also as approx. 2 years (first breeding with one year). However, the average age in the population directly after the breeding season including all fledglings (post-breeding population) amounts to just 0.5 years.

| | |
|---|---|
| average (total) life expectancy of <u>pre</u>-breeding population members, L | 3 years |

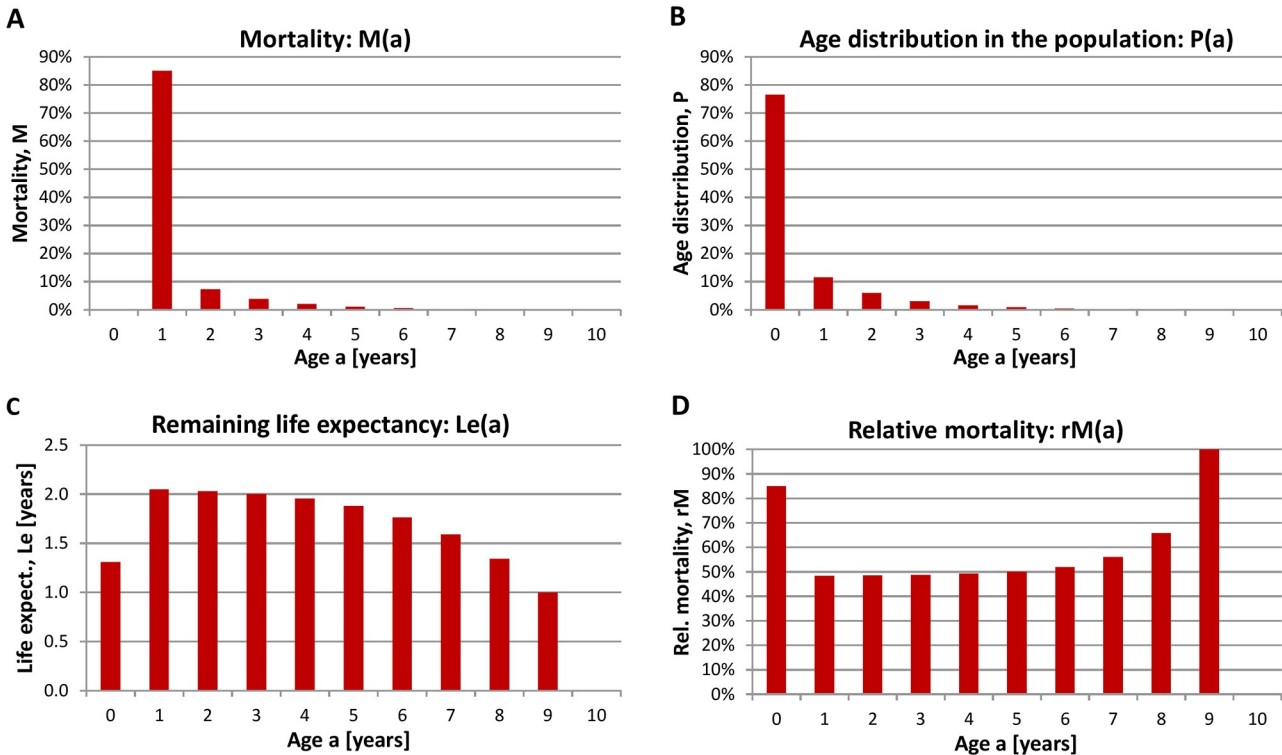

**Fig 5. Population dynamics of Black Redstarts in Halberstadt (Sachsen-Anhalt, Germany) based on data including all fledglings. (A)** Mortality M (a) (probability of dying at a given age), **(B)** age distribution in the post-breeding population (directly after fledging) P(a), **(C)** remaining life expectancy at a given age Le(a), and **(D)** relative mortality rM(a) (probability of dying when a certain age has been reached) of the Black Redstart. Calculations are based on age-differentiated bird counts taken from [15].

| | |
|---|---|
| average (total) life expectancy of fledgling (post-breeding population members), $L_f$ | 1.3 years |
| average age in the pre-breeding population, $A_{pre}$ | 2 years |
| average age in the post-breeding population, $A_{post}$ | 0.5 years |
| generation length, G<br>    2 years (first breeding with 1 year) | approx. |

Employing now the modified exponential mortality function as described in S1 File with the following input variables (portion of fledglings in the post-breeding population $g_f = 0.76$, mortality in the first year $M1 = 0.85$, population growth $r = 0$, maximum age $a_{max} = 10$ years, maximum age determined through ringing data as 10 years and 2 months [9]), the age-dependent mortality i.e. the probability of dying at a given age, the age distribution within the (post-breeding) population, the remaining life expectancy at a given age and the relative mortality i.e. the probability of dying when a certain age has been reached can be determined (Fig 5).

These results are quite different compared to those obtained for the other examples. The life expectancy is much shorter, particularly noticeable the high mortality in the first year and the resulting low life expectancy after fledging. Only birds which already reached their first birthday have an additional average life expectancy of two more years but their probability to die in the following year is also 50%. Only 0.8% of the Black Redstarts will get older than five years.

## Discussion

The mark-recapture method is certainly the most utilized method for the determination of avian population parameters. However, the method described here can complement the mark-recapture approach in particular if mark-recapture data are not existing. In the approach described here, age-specific results are derived from input data that are not age-specific (i.e. multi-year averages of the ratio of young to adult birds, population growth rate, as well as a single assumed or known maximum life span). These input data are then processed through an indirect deterministic exponential function to calculate the parameters of interest. In contrast to the mark-recapture approach which provides age-specific input data (and therefore estimates of mortality that actually reflect age-specific information) the method presented here creates age-specific results from lifetime averages.

Prerequisite for the approach described is that young and adult birds are easily distinguishable in the field. Additionally, this difference should be detectable in the field for a longer period of time or at least up to an age of the young birds at which they survived their more vulnerable periods of life. Moreover, there is also a need to have access to sufficiently large data sets. These data are becoming nowadays more frequently available in so called "Citizen Science" portals. Prerequisite for the applicability of these portals is that they are also paying attention to the collection of quantitative data. For example, the German portal "ornitho.de" pays special attention to determine age and sex differentiated bird numbers and also develops more tools for contributing data to e.g. the "International Waterbird Census" and other quantitative bird censuses. A potential insufficient accuracy of a single data set (one count of young and adult birds at a specific time and a specific place) collected by volunteers is balanced by the large number of data sets. For example, the determination of population parameters for the Whooper Swan is based on 14.923 single data sets collected in nine years (2012–2020) with a confidence interval of approx. 95% (see S2 File).

Furthermore, there might be other age-differentiated data sets collected for other purposes e.g. for the determination of breeding success that can now be easily re-evaluated using the described approach.

Of course, the method presented here but also the mark-recapture approach will not yield universal constants as, for example, the average life expectancy or the average first breeding age may change with changing conditions, e.g. climate or other changes affecting bird habitat and fitness. For example, when applied to birds living in captivity where abundant food, absence of predators, and even advanced medical care allows more birds to approach or reach their maximum possible age, numbers determined will be different from those determined for wild birds. Thus, all population parameters determined are only applicable to a defined population (e.g. time period and area).

The time period investigated should at least encompass several years to compensate for varying breeding success in different years. Also, the best time of the year for the census needs to be evaluated (see as examples the sections on the Black-headed Gull and also the Black Redstart).

Moreover, it is necessary to know if the total population considered is increasing, decreasing or stable in size. However, it should also be noted that the population parameters are not that sensitive to changes in the population size. For example, an erroneous assumption of a population increase ($r = 0.02$ corresponding to a doubling of the population in 35 years) will only change, for example, the determined average age in the population from $A = 7.4$ years ($g = 0.1$, constant population $r = 0$, $a_{max} = 30$ years) to $A = 7.3$ years ($g = 0.1$, increasing population $r = 0.02$, $a_{max} = 30$ years). However, the average individual life expectancy L depends

more strongly on a change in the population size than the average age in the population. Finally, the maximum age of the bird species needs to be known. But again, the population parameters are not that sensitive to changes of the maximum age. For example, a change in the maximum age in the population from 25 to 30 years would change the determined average age A in the population from 6.8 to 7.4 years (g = 0.1, r = 0). The most important input parameter, however, is the determined portion of young birds in the population which, considering an uncertainty of 5%, will affect all other results accordingly.

Another point that needs to be considered is the increased mortality of very young birds. Thus, counts are best carried out when the young birds survived the most dangerous period in their life. From this time on it can be assumed that the probability to die is approximately equal for young as well as for adult birds. This approach is applicable for Common Cranes and Whooper Swans which travel as families and where young birds benefit from the experience of the adults. On the other hand, comparative counts before breeding and after fledging will give additional information, for example on the enhanced mortality in the first year of a bird as shown in addition for the example of the Black Redstart.

A higher mortality in the first year affects the calculation more strongly as the portion of these birds in the population is larger than any other age group of adult birds. A higher mortality of birds in the first year through enhanced predation, bad weather and other environmental hazards is thus implemented in our approach. For all other age groups, we consider an equal average mortality. However, very old birds most likely have a higher average mortality and are more prone to die of bad weather conditions or other environmental hazards. But very old birds are only a minor part of the population and thus, their numbers do not affect the calculations strongly. If we would consider for all age groups different mortalities, we could not solve the equations with the input data of g = portion of young birds, r = growth of the population and $a_{max}$ = maximum known age. Of course this could be implemented in the calculations at the cost of more complexity and additional age-specific field data (see also the last section in the S1 File). The approach presented here uses indirect estimators and can be applied in circumstances where age-specific data are not or not easily accessible. Indirect estimations have been used before for the determination of other population parameters and proven their validity also in situations that otherwise could not be resolved, such as analysing survivorship of extinct animals from fossil data [17].

The best test for the validity of an indirect approach is agreement with the results obtained from direct methods, in our case from mark-recapture data. Unfortunately, we could not find published data concerning mortality and life expectancy for Common Cranes, Whooper Swans and Black Redstarts based on the mark-recapture method. However, we were able to find data on the mortality of Black-headed Gulls based on 351 ring recoveries [13]. Mortality determined by the method presented here and by using ring recoveries revealed—except for the first year mortality—a very good agreement. Thus, our approach to use age-differentiated bird counts for der determination of population parameters can nicely complement the mark-recapture approach or can be used at least for an approximation if mark-recapture data are not existing. The discrepancy for the first year mortality for the Black-headed Gulls was expectable as the high first year mortality was not considered in our calculations but incorporated in the analysis using mark-recapture data. Considering the high first year mortality in our approach requires age-differentiated bird counts in different time-segments starting with the first segment after fledging until age-differentiated bird counts reach constant values [11]. In most cases the high-first year mortality might be neglectable for first approximations but can be easily incorporated by simply using more age-differentiated bird counts from more time segments directly after fledging or even directly after hatching. Finally, it should be noted that the approach presented here might not only be useful for population parameter determinations

but could be also used for prediction; for example, for estimating the minimal long-term portion of young birds which is required to guarantee a stable population. And moreover, the method is not restricted to study bird populations, but can be applied to other species as well. Prerequisite in any case is the access to large data-sets of age-differentiated animal counts.

## Supporting information

**S1 File. Detailed description of the mathematical model background and instructions for calculations.**
(PDF)

**S2 File. Excel worksheet for the determination of population parameters including instructions for use and data sets for the determination of population parameters for the Common Crane, the Whooper Swan, the Black-headed Gull, and the Black Redstart.**
(XLSX)

**S3 File. The R file provided can be opened directly in R.** At the top of the file, parameters are given for the example of the Black-headed Gull (population growth "$r = 0$", first year mortality "$M1 = 0$", maximum age "$a_{max} = 30$", and first breeding age "$B1 = 3$"). The R file can be opened with any text editor and the parameters can be changed for other bird species.
(R)

**S4 File. Format of excel input file of counts of young and adult bird for calculations in R (counts of young and adult birds with the example of the Black-headed Gull).**
(XLSX)

## Acknowledgments

We are grateful to far more than 500 observers who reported more than 16,000 age-differentiated counts of Common Cranes, Whooper Swans and Black-Headed Gulls in *ornitho.de*. We are also grateful to Frank Hessing for his photo of Common Cranes and Bernd Nicolai for his photos of Black Redstarts and his comments on their molting pattern. And finally, we want to thank both reviewers who helped us to improve this manuscript.

## Author Contributions

**Conceptualization:** Werner Oldekop, Ursula Rinas.

**Data curation:** Werner Oldekop, Gerd Oldekop.

**Formal analysis:** Werner Oldekop.

**Methodology:** Werner Oldekop, Gerd Oldekop.

**Software:** Kai Vahldiek.

**Supervision:** Frank Klawonn.

**Validation:** Gerd Oldekop, Frank Klawonn.

**Visualization:** Gerd Oldekop.

**Writing – original draft:** Ursula Rinas.

**Writing – review & editing:** Ursula Rinas.

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
