## [Decision Letter · Decision Letter 0]

21 Jun 2022

PONE-D-22-11340Counting young birds: a simple tool for the determination of avian population parametersPLOS ONE

Dear Dr. Rinas,

Thank you for submitting your manuscript to PLOS ONE. After careful consideration, we feel that it has merit but does not fully meet PLOS ONE’s publication criteria as it currently stands. Therefore, we invite you to submit a revised version of the manuscript that addresses the points raised during the review process.

We look forward to receiving your revised manuscript.

Kind regards,

Maria Andreína Pacheco, Ph.D.

Academic Editor

PLOS ONE

Journal Requirements:

Additional Editor Comments:

My apologies for the delay to send you the response.

Reviewers' comments:

Reviewer's Responses to Questions

**Comments to the Author**

1. Is the manuscript technically sound, and do the data support the conclusions?

Reviewer #1: Yes

Reviewer #2: Yes

2. Has the statistical analysis been performed appropriately and rigorously? 

Reviewer #1: Yes

Reviewer #2: No

3. Have the authors made all data underlying the findings in their manuscript fully available?

Reviewer #1: Yes

Reviewer #2: Yes

4. Is the manuscript presented in an intelligible fashion and written in standard English?

Reviewer #1: Yes

Reviewer #2: Yes

5. Review Comments to the Author

Reviewer #1: The manuscript titled “Counting young birds: a simple tool for the determination of avian population parameters” is an interesting approach to the issue of obtaining valuable demographic parameters from limited count data. The basic premise of using age-differentiated count data rather than mark-recapture data to calculate things like age-specific life expectancy or the age distribution of a population is potentially very useful for conservation managers.

I did not have any concerns about the actual mathematical approach, but I do provide several general comments about the manuscript. Specifically, I was concerned about how the approach is laid out and how it is differentiated from other methods to obtain similar results. There were also a few aspects of the data used in the analysis that I believe the authors may want to expand upon in their discussion. I describe those general comments first, and then end with several minor comments about specific lines of text in the manuscript that could be improved.

The general analytical approach laid out in the manuscript seems good, however there are important limitations to the method that should be more clearly explained, particularly in the Abstract and Introduction. Most importantly, if I understand correctly, the major age-specific results displayed in the manuscript’s figures are derived from a few inputs that are not age-specific (i.e., multi-year averages of the ratio of young:adult birds and population growth rate, as well as a single assumed maximum life span), which are then run through a deterministic exponential function to calculate the parameters of interest. The authors compare their method to a mark-recapture approach, but a mark-recapture approach provides age-specific data (and therefore estimates of mortality that actually reflect age-specific information), whereas the authors’ method creates age-specific results from lifetime averages. The authors allude to the fact that a mark-recapture approach is different in the first line of the discussion but are vague about exactly how. I believe that many readers will see the age-specific results in the figures without appreciating that the differences among years are not a result of data, but rather simply reflect output from a deterministic function. Of course, this does not mean that the authors’ method does not provide useful results (especially if age-specific data are not available), but I feel that there should be more detail—certainly in the discussion, but perhaps also in the Introduction—describing how the results from this method use a function to calculate estimates of these age-specific demographic parameters, rather than obtaining such estimates from age-specific data.

When reading the manuscript, one initial question I had was: why are the majority of the methods summarized in the manuscript on L70 as “some additional mathematics”, and then laid out in detail in a supplement, rather than included in the main manuscript as a methods section? I suspect that this may relate to the description of the manuscript as a “communication” on L 42, perhaps suggesting that the manuscript was originally intended as a shorter-format paper in a different journal. I recommend moving much of S1 to a methods section in the manuscript, perhaps leaving some ancillary sections (such as 3.2.2 through 3.2.3, 3.3.1 through 3.3.2, and 3.4) in a supplement.

Within those methods (S1), I struggled to determine what part of the material came from existing literature on more traditional life table calculations, and what part came from novel work by the authors. Part of my struggle could have been due to differences in the notation. It may be helpful for the authors to cite some existing literature for steps/equations that were known prior to this work (and perhaps explain when notation differs from those sources), and then explicitly describe which steps in the mathematical logic were developed for this manuscript. This is important, I believe, because some of the equations are difficult to understand based solely on those that come before, without also having access to the more comprehensive theory of life table mathematics.

My final general comments concern the nature of the bird data that the authors suggest may be used with their methods by others, particularly the use of the term “citizen science”. I am not very familiar with the protocols for ornitho.de data, however citizen science programs are extremely variable in both precision and accuracy of bird count data, largely contingent on what effort data are collected, the amount of volunteer training, and the procedures for data verification. Observers will almost invariably have less-than-perfect detection for birds they are counting (although that applies to professional researchers as well). The current manuscript would benefit from at least a discussion of potential bias from differences in detection of young vs. adult birds, as uncertainty in that ratio was mentioned as having an important effect on the results. Even if the counts of species used as examples here are believed to have relatively little difference in the detection of young vs. adult birds, such differences may limit the use of this method for other species or with data from other citizen science programs.

Minor comments:

L39: The phrase “insects up to mammals” implies a natural hierarchy of importance among animals, which may exacerbate existing biases for the study of charismatic species. I recommend changing the sentence to read: “…can be marked, such as insects and mammals.”

L46: This is the first use of the word “respectively” in the text, which I believe the authors are using incorrectly. I believe what they intended to mean with the word is “that is” (or “i.e.”).

L58-66: Although I did appreciate the analogy of a tank of water, it seemed odd to introduce the analogy before simply explaining the relationship in direct terms. It is also simple enough that I don’t know if the analogy is actually necessary (perhaps a diagram could do the same job?). I think that many biologists reading that paragraph will, as I first did, begin to think of all the reasons why a bird population will not behave as a tank of water, even though that is one of the points of the approach: to make several simplifying assumptions about the birds, allowing us to estimate the demographic parameters with only age-ratio data.

S1; 3.0: The authors state that “…predation, shooting, extreme weather, accidents and infections. These causes usually strike the birds independent of their age, so that each year a certain proportion of birds dies from all age groups.” I would disagree; in many species (likely including the example species), susceptibility to all of these factors could easily be dependent on a bird’s age. However, most of those factors will likely increase in importance as a bird ages, so as long as the exponential model provides a good fit to the data, the authors’ approach should still be appropriate. This is probably one of the biggest reasons why, as the authors suggest, their method is not able to completely replace a mark-recapture study, which could provide data to address those factors.

Reviewer #2: The proposed method is creative, novel, and potentially applicable in many other studies and situations. The example of the water tank is simple but pedagogic. I found no problems either in the system of equations or the rationale behind it. However, some aspects can be improved. In conclusion, the study is worth to be published after some major changes.

INTRODUCTION

1) The authors should consider modifying the manuscript to target a broader audience. For example, regarding the mark/recapture method, they simply mention that “Although this method is certainly the gold standard it has also some drawbacks. Most obviously, it depends on animal marking and their recaptures/resightings” (lines 39-41) which is not necessarily an evident problem for many readers.

2) The authors simply mention the “citizen science databases”, without explaining what these databases are or citing examples where they have been successfully applied. Both aspects, including the drawbacks associated with this source of information, must be commented/discussed at some point.

3) The authors mention that “The main prerequisites of this method encompass that young and adult birds are easily distinguishable in the field and the existence of large data sets” (lines 39-41). However, and keeping in mind the water tank example (and their statemen in S1 that “counting year is usually not the first of January but the month when migration starts”), a question comes in mind: can this method be applied to tropical species (that is, to the largest bulk of the bird biodiversity) with much less synchronized reproductive events?

4) Kindly remove “Sweden and Finland” (line 94).

5) Kindly include composed photos or pictures illustrating the age-related differences of each one of the species studied.

MATHEMATICAL BACKGROUND

6) The authors include some confidence measurements in the S2, but these are omitted in the main text. Confidence intervals are important to avoid imprecisions such as “there is also need to have access to sufficiently large data sets” (lines 285-286) or “the time period investigated should encompass several years” (lines 298-299). At this point, it’s impossible to know what “sufficiently large” or “several years” represents with precision.

7) Mortality may be affected by many different factors across species, for which I would expect differences in the probability distributions of the respective mortalities among the species studied (and, of course, divergencies from the modified geometric one used in the present study). Considering this and given that (a) the inclusion of confidence intervals improves the life tables and (b) the mathematics involved in the present study are not complicated, I would strongly suggest using bootstrap to approximate the shape of the sampling distribution and calculate the tables at each run. This approach is computationally more intensive, but the modern computers and the widely available resources, such as Python or R, should be easily up to the task.

DISCUSSION

8) The authors highlight the limitations regarding the information available measured in the field for the species considered in the study (lines 324-325). However, they could enrich the discussion by comparing their results against other studies on phylogenetically related species of cranes, swans, gulls and chats.

9) The authors should highlight the validity and relevance of using indirect estimators for populations parameters by discussing how this approach has successfully been used in other studies, including situations that otherwise could not be resolved, such as fossils (https://doi.org/10.1139/E10-051)

10) Figures should be improved, by reducing the space among plots and highlight the letters and terms referring to some parameter.

SUPPLEMENTARY MATERIALS

11) Please, kindly highlight the letters referring to parameters in the text (by italicizing them or using an alternative font). Sentences such as “The residual life expectancy of a bird that has reach already age a…” can be misleading.

12) In the previously cited sentence, please change “has reach” by “has reached”.

13) Please, clearly define the formula terms the first time they appear. For example, “k” in formula F8.

6. PLOS authors have the option to publish the peer review history of their article (what does this mean?). If published, this will include your full peer review and any attached files.

Reviewer #1: **Yes: **Michael Schrimpf

Reviewer #2: No

---

## [Author Response · Author response to Decision Letter 0]

21 Nov 2022

Many thanks to the reviewers for useful comments helping us to improve the manuscript. The response to the comments is included in red.

Moreover, all changes in the revised version of the manuscript are indicated in red.

Submission ID [PONE-D-22-11340] - [EMID:bcb6167ad3dbac2f]

Reviewer #1: 

The manuscript titled “Counting young birds: a simple tool for the determination of avian population parameters” is an interesting approach to the issue of obtaining valuable demographic parameters from limited count data. The basic premise of using age-differentiated count data rather than mark-recapture data to calculate things like age-specific life expectancy or the age distribution of a population is potentially very useful for conservation managers.

I did not have any concerns about the actual mathematical approach, but I do provide several general comments about the manuscript. Specifically, I was concerned about how the approach is laid out and how it is differentiated from other methods to obtain similar results. There were also a few aspects of the data used in the analysis that I believe the authors may want to expand upon in their discussion. I describe those general comments first, and then end with several minor comments about specific lines of text in the manuscript that could be improved.

Thank you very much for these encouraging comments.

The general analytical approach laid out in the manuscript seems good, however there are important limitations to the method that should be more clearly explained, particularly in the Abstract and Introduction. Most importantly, if I understand correctly, the major age-specific results displayed in the manuscript’s figures are derived from a few inputs that are not age-specific (i.e., multi-year averages of the ratio of young: adult birds and population growth rate, as well as a single assumed maximum life span), which are then run through a deterministic exponential function to calculate the parameters of interest. The authors compare their method to a mark-recapture approach, but a mark-recapture approach provides age-specific data (and therefore estimates of mortality that actually reflect age-specific information), whereas the authors’ method creates age-specific results from lifetime averages. The authors allude to the fact that a mark-recapture approach is different in the first line of the discussion but are vague about exactly how. I believe that many readers will see the age-specific results in the figures without appreciating that the differences among years are not a result of data, but rather simply reflect output from a deterministic function. Of course, this does not mean that the authors’ method does not provide useful results (especially if age-specific data are not available), but I feel that there should be more detail—certainly in the discussion, but perhaps also in the Introduction—describing how the results from this method use a function to calculate estimates of these age-specific demographic parameters, rather than obtaining such estimates from age-specific data.

This general comment is certainly true and very important! We include and discuss this point more thoroughly in the Discussion section of the revised manuscript and also mention this point in the Abstract of the revised manuscript. Thank you for directing us to this point! 

When reading the manuscript, one initial question I had was: why are the majority of the methods summarized in the manuscript on L70 as “some additional mathematics”, and then laid out in detail in a supplement, rather than included in the main manuscript as a methods section? I suspect that this may relate to the description of the manuscript as a “communication” on L 42, perhaps suggesting that the manuscript was originally intended as a shorter-format paper in a different journal. I recommend moving much of S1 to a methods section in the manuscript, perhaps leaving some ancillary sections (such as 3.2.2 through 3.2.3, 3.3.1 through 3.3.2, and 3.4) in a supplement.

We have carefully considered this comment but do not feel comfortable to split the mathematics into a section that is included in the main manuscript and another section found in the Supporting information. Even with splitting the mathematics would be a very large section in the main manuscript. We feel that it is better to present the results of the calculations in the main manuscript and address those readers to the Supporting information which are interested in the mathematical background and which are interested to perform these calculations themselves with other bird count data.

Within those methods (S1), I struggled to determine what part of the material came from existing literature on more traditional life table calculations, and what part came from novel work by the authors. Part of my struggle could have been due to differences in the notation. It may be helpful for the authors to cite some existing literature for steps/equations that were known prior to this work (and perhaps explain when notation differs from those sources), and then explicitly describe which steps in the mathematical logic were developed for this manuscript. This is important, I believe, because some of the equations are difficult to understand based solely on those that come before, without also having access to the more comprehensive theory of life table mathematics.

The method described is based on a “chemical/reactor engineering approach” mainly using mass balances and kinetic equations. We treat birds as molecules/particles which are put as young birds (inflow of particles into a tank) into a system (bird population = a tank) where they transform into adult birds, leaving the system is considered as death (outflow). We have tried to clarify this approach more clearly in the revised version of the manuscript.

My final general comments concern the nature of the bird data that the authors suggest may be used with their methods by others, particularly the use of the term “citizen science”. I am not very familiar with the protocols for ornitho.de data, however citizen science programs are extremely variable in both precision and accuracy of bird count data, largely contingent on what effort data are collected, the amount of volunteer training, and the procedures for data verification. Observers will almost invariably have less-than-perfect detection for birds they are counting (although that applies to professional researchers as well). The current manuscript would benefit from at least a discussion of potential bias from differences in detection of young vs. adult birds, as uncertainty in that ratio was mentioned as having an important effect on the results. Even if the counts of species used as examples here are believed to have relatively little difference in the detection of young vs. adult birds, such differences may limit the use of this method for other species or with data from other citizen science programs.

We have discussed this point more thoroughly in the Discussion section. 

Minor comments:

L39: The phrase “insects up to mammals” implies a natural hierarchy of importance among animals, which may exacerbate existing biases for the study of charismatic species. I recommend changing the sentence to read: “…can be marked, such as insects and mammals.”

We have changed the wording. It was not our intention to strengthen a negative bias against certain animals but to point out that the method can be used for all kinds of animals (even small ones such as insects) if species specific markings are available.

L46: This is the first use of the word “respectively” in the text, which I believe the authors are using incorrectly. I believe what they intended to mean with the word is “that is” (or “i.e.”).

It is now corrected.

L58-66: Although I did appreciate the analogy of a tank of water, it seemed odd to introduce the analogy before simply explaining the relationship in direct terms. It is also simple enough that I don’t know if the analogy is actually necessary (perhaps a diagram could do the same job?). I think that many biologists reading that paragraph will, as I first did, begin to think of all the reasons why a bird population will not behave as a tank of water, even though that is one of the points of the approach: to make several simplifying assumptions about the birds, allowing us to estimate the demographic parameters with only age-ratio data.

We used this analogy as we (or some of us) are chemical engineers and are used to calculate the “aging” of molecules in reaction tanks. We thought that this simple analogy is helpful for those readers which are not so familiar with mathematics. We kept this analogy in the revised manuscript but introduced the idea of “coloured particles” as birds to make it more descriptive and better understandable.

S1; 3.0: The authors state that “…predation, shooting, extreme weather, accidents and infections. These causes usually strike the birds independent of their age, so that each year a certain proportion of birds dies from all age groups.” I would disagree; in many species (likely including the example species), susceptibility to all of these factors could easily be dependent on a bird’s age. However, most of those factors will likely increase in importance as a bird ages, so as long as the exponential model provides a good fit to the data, the authors’ approach should still be appropriate. This is probably one of the biggest reasons why, as the authors suggest, their method is not able to completely replace a mark-recapture study, which could provide data to address those factors.

We have clarified this point in the Discussion section. A higher mortality is considered in our calculation for young birds (higher mortality in the first year) in the example of the Black Redstarts. A higher mortality in the first year affects the calculation more strongly as the portion of these birds in the population is larger than any other age group of adult birds. A higher mortality of birds in the first year through enhanced predation, bad weather and other environmental hazards is thus implemented in our approach. If we consider for all other age groups also different mortalities, we can not solve the equations with the input data of g=portion of young birds, r=growth of the population and amax=maximum known age. Of course this could be implemented in the calculations at the cost of more complexity and additional age-specific field data (see also the last section in the Supporting Information S1: 3.5 Other approaches and further perspectives). Again, we consider our approach as a reasonable simple approach complementing but not replacing the mark-recapture method. 

Reviewer #2:

The proposed method is creative, novel, and potentially applicable in many other studies and situations. The example of the water tank is simple but pedagogic. I found no problems either in the system of equations or the rationale behind it. However, some aspects can be improved. In conclusion, the study is worth to be published after some major changes.

Thank you very much for these encouraging comments.

INTRODUCTION

1) The authors should consider modifying the manuscript to target a broader audience. For example, regarding the mark/recapture method, they simply mention that “Although this method is certainly the gold standard it has also some drawbacks. Most obviously, it depends on animal marking and their recaptures/resightings” (lines 39-41) which is not necessarily an evident problem for many readers.

We have extended the discussion regarding this point in the introduction.

2) The authors simply mention the “citizen science databases”, without explaining what these databases are or citing examples where they have been successfully applied. Both aspects, including the drawbacks associated with this source of information, must be commented/discussed at some point.

We have discussed this point more thoroughly in the Discussion part of revised version of the manuscript. Our data were all taken from the german database “orrnitho.de” which pays special attention to age and sex differentiated bird counts. For example, this database is now also utilzed to contribute data to the “International Waterbird Census” and other quantitative Birds censuses.

3) The authors mention that “The main prerequisites of this method encompass that young and adult birds are easily distinguishable in the field and the existence of large data sets” (lines 39-41). However, and keeping in mind the water tank example (and their statemen in S1 that “counting year is usually not the first of January but the month when migration starts”), a question comes in mind: can this method be applied to tropical species (that is, to the largest bulk of the bird biodiversity) with much less synchronized reproductive events?

This is an interesting point which we have not considered. If breeding is not or less synchronized our approch should be even easier to adapt. However, we believe that for each specific bird species the circumstances have to be evaluated.

4) Kindly remove “Sweden and Finland” (line 94).

We prefer to keep Sweden and Finland as e.g. Norway is also a Scandinavian Country but Norway has no important breeding population of Common Cranes.

5) Kindly include composed photos or pictures illustrating the age-related differences of each one of the species studied.

We have included now for all species photos of young and adult birds pointing to the differences between both of them.

MATHEMATICAL BACKGROUND

6) The authors include some confidence measurements in the S2, but these are omitted in the main text. Confidence intervals are important to avoid imprecisions such as “there is also need to have access to sufficiently large data sets” (lines 285-286) or “the time period investigated should encompass several years” (lines 298-299). At this point, it’s impossible to know what “sufficiently large” or “several years” represents with precision.

We have now mentioned confidence intervals also in the main manuscript.

7) Mortality may be affected by many different factors across species, for which I would expect differences in the probability distributions of the respective mortalities among the species studied (and, of course, divergencies from the modified geometric one used in the present study). Considering this and given that (a) the inclusion of confidence intervals improves the life tables and (b) the mathematics involved in the present study are not complicated, I would strongly suggest using bootstrap to approximate the shape of the sampling distribution and calculate the tables at each run. This approach is computationally more intensive, but the modern computers and the widely available resources, such as Python or R, should be easily up to the task.

We have now implemented a tool to calculate the main parameters using R providing confidence intervals based on 10 000 bootstrap simulations. Kai Vahldiek who implemented the R-Code is now also included in the author list. 

DISCUSSION

8) The authors highlight the limitations regarding the information available measured in the field for the species considered in the study (lines 324-325). However, they could enrich the discussion by comparing their results against other studies on phylogenetically related species of cranes, swans, gulls and chats.

We have discussed the results obtained by our approach for the Black-Headed Gull to data obtained by the mark-recapture method. 

9) The authors should highlight the validity and relevance of using indirect estimators for populations parameters by discussing how this approach has successfully been used in other studies, including situations that otherwise could not be resolved, such as fossils (https://doi.org/10.1139/E10-051)

We have extended our discussion on utilizing indirect estimations for population parameter estimations.

10) Figures should be improved, by reducing the space among plots and highlight the letters and terms referring to some parameter.

We have reduced the space among the graphs. Letters and terms are given in the Figure and also explained in the Figure captions.

SUPPLEMENTARY MATERIALS

11) Please, kindly highlight the letters referring to parameters in the text (by italicizing them or using an alternative font). Sentences such as “The residual life expectancy of a bird that has reach already age a…” can be misleading.

We have now utilized bold letters in brakets for the parameters in the text (main manuscript and Supporting Information). 

12) In the previously cited sentence, please change “has reach” by “has reached”.

This is corrected in the revised version of the Supporting Information.

13) Please, clearly define the formula terms the first time they appear. For example, “k” in formula F8.

This has been changed in the revised version as suggested by this reviewer.

---

## [Decision Letter · Decision Letter 1]

19 Dec 2022

Counting young birds: a simple tool for the determination of avian population parameters

PONE-D-22-11340R1

Dear Dr. Rinas,

We’re pleased to inform you that your manuscript has been judged scientifically suitable for publication and will be formally accepted for publication once it meets all outstanding technical requirements.

Kind regards,

M. Andreína Pacheco, Ph.D.

Academic Editor

PLOS ONE

Reviewers' comments:

Reviewer's Responses to Questions

**Comments to the Author**

1. If the authors have adequately addressed your comments raised in a previous round of review and you feel that this manuscript is now acceptable for publication, you may indicate that here to bypass the “Comments to the Author” section, enter your conflict of interest statement in the “Confidential to Editor” section, and submit your "Accept" recommendation.

Reviewer #2: All comments have been addressed

2. Is the manuscript technically sound, and do the data support the conclusions?

Reviewer #2: Yes

3. Has the statistical analysis been performed appropriately and rigorously? 

Reviewer #2: Yes

4. Have the authors made all data underlying the findings in their manuscript fully available?

Reviewer #2: Yes

5. Is the manuscript presented in an intelligible fashion and written in standard English?

Reviewer #2: Yes

6. Review Comments to the Author

Reviewer #2: After carefully reading this second version of the manuscript "Counting young birds: a simple tool for the determination of avian population parameters. (PONE-D-22-11340R1) as well as the authors' responses to my comments, I conclude that the authors have adequately responded to my observations to the previous version, so I consider that this manuscript is now publishable.

I thank the authors for having included all the observations that I consider fundamental and for having satisfactorily explained to me those that were not included in this version.

7. PLOS authors have the option to publish the peer review history of their article (what does this mean?). If published, this will include your full peer review and any attached files.

Reviewer #2: **Yes: **Paolo Ramoni Perazzi

---

## [Editor Report · Acceptance letter]

9 Feb 2023

PONE-D-22-11340R1 

Counting young birds: a simple tool for the determination of avian population parameters 

Dear Dr. Rinas:

I'm pleased to inform you that your manuscript has been deemed suitable for publication in PLOS ONE. Congratulations! Your manuscript is now with our production department. 

Kind regards, 

on behalf of

Dr. M. Andreína Pacheco 

Academic Editor

PLOS ONE